# Nuclear Magnetic Resonance-Measured Ionized Magnesium Is Inversely Associated with Type 2 Diabetes in the Insulin Resistance Atherosclerosis Study

**DOI:** 10.3390/nu14091792

**Published:** 2022-04-25

**Authors:** Erwin Garcia, Irina Shalaurova, Steven P. Matyus, Joelle C. Schutten, Stephan J. L. Bakker, Robin P. F. Dullaart, Margery A. Connelly

**Affiliations:** 1Laboratory Corporation of America Holdings (Labcorp), Morrisville, NC 27560, USA; garce14@labcorp.com (E.G.); shalaui@labcorp.com (I.S.); matyuss@labcorp.com (S.P.M.); 2Department of Internal Medicine, Division of Nephrology, University Medical Center Groningen, University of Groningen, 9700 RB Groningen, The Netherlands; j.c.schutten@umcg.nl (J.C.S.); s.j.l.bakker@umcg.nl (S.J.L.B.); 3Department of Internal Medicine, Division of Endocrinology, University Medical Center Groningen, University of Groningen, 9700 RB Groningen, The Netherlands; dull.fam@12move.nl

**Keywords:** ionized magnesium, nuclear magnetic resonance spectroscopy, type 2 diabetes

## Abstract

The aims were to optimize a nuclear magnetic resonance (NMR)-based assay for quantifying ionized or free magnesium and investigate its association with type 2 diabetes (T2D). A high-throughput, ionized magnesium assay was optimized and evaluated. Plasma magnesium was quantified, and associations with T2D were ascertained in Insulin Resistance Atherosclerosis Study (IRAS) participants. Coefficients of variation for the ionized magnesium assay ranged from 0.7–1.5% for intra-assay and 4.2–4.7% for inter-assay precision. In IRAS (n = 1342), ionized magnesium was significantly lower in subjects with prediabetes and T2D than in normoglycemic subjects, and lower in participants with T2D than those with prediabetes (*p* < 0.0001). Cross-sectional regression analyses revealed that magnesium was associated with T2D at baseline in models adjusted for multiple clinical risk factors (*p* = 0.032). This association appeared to be modified by sex, in such a way that the associations were present in women (OR = 0.54 (95% CI 0.37–0.79), *p* = 0.0015) and not in men (OR = 0.98 (95% CI 0.71–1.35), *p* = 0.90). Longitudinal regression analyses revealed an inverse association between magnesium and future T2D in the total population (*p* = 0.035) that was attenuated by LP-IR (*p* = 0.22). No interactions were detected between magnesium and age, race, BMI, glucose, insulin, triglycerides, or LPIR for the prospective association with future T2D. However, a significant interaction between magnesium and sex was present, now with a trend for an association in men (OR = 0.75 (95% CI 0.55–1.02), *p* = 0.065 and absence of an association in women (OR = 1.01 (0.76–1.33), *p* = 0.97). Conclusions: lower ionized magnesium, as measured by an NMR-based assay optimized for accuracy and precision, was associated cross-sectionally with T2D at baseline and longitudinally with incident T2D in IRAS.

## 1. Introduction

Along with sodium, potassium, and calcium, magnesium is critically important for human physiology and metabolism [1,2,3,4,5]. As such, magnesium has been identified as a cofactor for over 600 enzymatic reactions [5,6,7,8]. The majority of total body magnesium is located in the bone; however, it can also be found in muscle and other soft tissues [1,2,3,4]. Circulating magnesium accounts for about 1–3% of total body magnesium and can be divided into three fractions: (1) complexed with anions (5–15%), (2) bound to protein (20–30%), and (3) ionized or free (55–70%) [1,2,3,4]. In healthy individuals, total serum magnesium concentration, measured using a chemistry-based assay, is tightly regulated in the physiological range of 0.70–1.05 mM [9,10] via a balance between intestinal absorption, renal excretion, and absorption and desorption from tissues such as the bone, with renal excretion playing the primary role in maintaining magnesium homeostasis [1,2,3,4,5]. Low circulating total serum magnesium, or hypomagnesemia (defined as less than 0.70 mM), has been shown to be associated with hyperglycemia, insulin resistance, dyslipidemia, hypertension, migraine, epilepsy, and Alzheimer’s disease, as well as an increased risk of metabolic syndrome, type 2 diabetes (T2D), chronic kidney disease, and cardiovascular (CV) disease (e.g., stroke, myocardial infarction, and arrhythmias) [1,2,3,4,5,11,12,13,14].

Total magnesium levels, however, do not necessarily represent the amount of magnesium that is freely available for biological reactions. Ionized or free magnesium, on the other hand, is considered to be the more bioactive form of magnesium [1,2,3,15]. The normal range for serum ionized magnesium is 0.55–0.75 mM [1]. Given the observed associations between whole blood, serum and/or plasma levels of magnesium and multiple chronic diseases, a simple and reliable assessment of circulating bioactive magnesium concentration is important. To this end, a high-throughput nuclear magnetic resonance (NMR)-based assay for determining plasma ionized or free magnesium concentrations in the clinical laboratory was developed [11]. A significant benefit of this assay is that it allows for the simultaneous quantification of lipoprotein particles, small molecule metabolites, GlycA, and the Lipoprotein Insulin Resistance Index (LP-IR) [16,17,18,19,20,21,22,23,24]. GlycA is the name given to the NMR signal that arises from circulating acute phase proteins, and as such, is an NMR-specific marker of systemic inflammation [19]. GlycA has been shown to be associated with CV disease and T2D [25,26,27]. LP-IR is a lipoprotein-based assay for assessing a patient’s insulin resistance status [16]. High LP-IR scores have also been shown to be associated with incident T2D [28]. It was reported that plasma magnesium levels, determined using this NMR-based assay, are associated with risk of developing T2D in a large general population-based cohort [11]. The aims of the current study were to: (1) improve the precision and accuracy of this magnesium assay by assay optimization, (2) assess assay performance, and (3) determine the longitudinal association of NMR-measured plasma ionized magnesium with future T2D in the Insulin Resistance Atherosclerosis Study, a diverse population of higher risk individuals, including individuals with prediabetes.

## 2. Materials and Methods

### 2.1. Optimization of an NMR-Based Ionized or Free Magnesium Assay

Ethylenediaminetetraacetic acid (EDTA) plasma samples were diluted 1:1 with phosphate buffer (pH 7.4) containing 5 mM EDTA, as previously described for the acquisition of NMR spectra for the NMR LipoProfile^®^ test [17]. EDTA in the buffer ensures complete chelation of free ionized magnesium present in plasma, as well as magnesium that may not be tightly bound to citrates, phosphates, or proteins. Hence, it is suggested that EDTA plasma is the preferred specimen for NMR-based ionized magnesium quantification. Proton NMR spectra were collected on 400 MHz Vantera^®^ Clinical Analyzers (Labcorp, Morrisville, NC, USA) at 47 °C as previously described [17,18,22]. The NMR spectra were deconvoluted using a proprietary algorithm to quantify chelatable, free, and ionized magnesium as previously described [11]. Briefly, the singlet signal emanating from 4 protons within the ethylene moiety in the magnesium-EDTA complex, with a peak at 2.66 ppm in the NMR spectrum, was used for quantification (Figure 1). Because the magnesium-EDTA signal overlapped with protein signals, a 16 Hz wide region of the peak was integrated including components in the deconvolution that account for the protein signal encompassing approximately 50 Hz. The relationship between the magnesium-EDTA signal peak area and magnesium concentrations was established by spiking experiments using dialyzed plasma, and the conversion factor was applied to translate the magnesium-EDTA signal area to a concentration expressed in µM. To optimize the previously published magnesium assay, several post spectral acquisition features were implemented, including: (1) automatic phasing of each NMR spectrum, (2) correction factors that minimize changes in shimming and pH that may occur during sample testing, and (3) Becton Dickinson gel barrier tube detection and rejection. The latter is based on the level of a substance that elutes from the gel when samples are left on the gel too long and the signal of which interferes with the ability of the software to quantify magnesium properly.

Analytical performance and clinical validation of GlycA and the Lipoprotein Insulin Resistance Index (LP-IR) (0–100; from least (0) to most (100) insulin resistant) have been previously reported [16,19,26,28].

### 2.2. Evaluation of Assay Performance

A Slide-A-Lyzer dialysis 10 kDa molecular weight cutoff cassette (Thermo Scientific, Rockford, IL, USA) was used to dialyze and deplete EDTA plasma samples of free magnesium for determining the limits of blank (LOB). EDTA plasma pools (n = 5) containing extremely low concentrations of magnesium were tested to determine the limits of detection (LOD) and quantitation (LOQ) according to Clinical and Laboratory Standards Institute (CLSI) guidelines [29]. Accuracy was determined through recovery experiments. Linearity was tested in quadruplicate across the biological range from 15.2–2715 µM by preparing dialyzed EDTA plasma samples and spiking with known magnesium concentrations. Precision (within-run or intra-assay and within-laboratory or inter-assay) was determined based on CLSI guidelines using EDTA plasma pools targeted at low and high concentrations of magnesium [30]. Mean concentration, standard deviation, and coefficients of variation (%CV) were calculated for each pool. To assess analyte stability, EDTA plasma specimens from 10 donors were collected. Stability was assessed at ambient room (20–25 °C), refrigerated (2–8 °C), and frozen (−20 °C and <−70 °C) temperatures, as well as with several cycles of freezing and thawing. To determine long-term stability in samples frozen at <−70 °C (9 years), spectra were collected for 563 samples on a Vantera in 2012, and the same samples were tested again in 2021. The digitized and stored spectra were reanalyzed using the optimized magnesium assay and results were compared. Mean results for all donors were evaluated with acceptable differences falling within ±10% of the day 0 (draw day) mean.

### 2.3. Reference Interval Determination

A study was conducted to determine the reference interval for ionized magnesium in EDTA plasma from 567 apparently healthy adult men and women between the ages 18 and 84 years. Samples were drawn from fasting and non-fasting volunteers into purple top EDTA plasma tubes. The study population is described in detail in [17]. All volunteer donors provided informed consent and the study was approved by an Institutional Review Board (IRB), Chesapeake Research Review, Inc. (Raleigh, NC, USA), in 2012 (MOD00049155). Magnesium was measured using the NMR assay, as described above, in singlicate. Samples with missing magnesium results were excluded for further analysis, leaving a total of 564 participants. The reference interval was determined from the 2.5th and 97.5th percentiles.

### 2.4. Method Comparison Study

The method comparison study was conducted within CLSI Guidelines. The reference method used was the Roche colorimetric total magnesium assay; a colorimetric assay that measures total magnesium in serum or heparin plasma. The method is based on the chemical reaction of magnesium with xylidyl blue in an alkaline solution containing ethylene glycol-bis(β-aminoethyl ether)-N,N,N′,N′-tetraacetic acid (EGTA) in order to mask endogenous calcium. In the solution, magnesium forms a purple complex with the xylidyl blue diazonium salt, and the concentration of magnesium is determined photometrically via the decrease in the xylidyl blue absorbance (505/600 nm). Magnesium results were generated in both the Roche and NMR assays using plasma samples from the Prevention of End-Stage Renal Disease (PREVEND) study (n = 5040). Information about this study was previously reported and are described briefly below [11]. Deming and Passing–Bablok regression was used to compare the results produced by the Roche colorimetric-based total magnesium and NMR-based ionized magnesium assays. Bland–Altman plots were created to evaluate potential differences and outliers.

The PREVEND study was an observational prospective cohort study which was set up to investigate the predictive value of albuminuria in relation to renal disease and CV outcomes in the general population in the northern part of the Netherlands (NL). The study design and recruitment procedures were described in detail previously [11]. Briefly, non-pregnant participants (aged 28–75 years) free of type 1 diabetes were selected from the population of the city of Groningen, NL to create a study population with varying age and urine albumin levels. Baseline measurements were performed in 8592 participants between 1997 and 1998. For the current study, NMR results were collected for EDTA plasma samples collected during the second screening round (2001–2003) which included 6894 subjects. Subjects used for this study included participants from the second screening round with both Roche and NMR measured magnesium results. The PREVEND study was approved by the medical ethics committee of the University Medical Center Groningen and conducted in accordance with the Declaration of Helsinki ethical approval code METC 96/01/022 (NCT03073018l). All participants provided informed consent.

### 2.5. Insulin Resistance Atherosclerosis Study (IRAS)

The IRAS cohort comprised 1625 participants from 4 clinical centers across the United States between October 1992 and April 1994. Subjects included men and women aged 40–69 years from three different ethnic groups: Hispanic, African American, and non-Hispanic white. Details of the study population, research methods, and exclusion criteria were published in [31]. Diabetes was defined using World Health Organization (WHO) criteria: fasting glucose concentration ≥ 7.0 mmol/L and/or 2 h glucose concentration ≥ 11.1 mmol/L by a 75 g oral glucose tolerance test (OGTT) [32]. The use of insulin during a 5 year period before entry was an exclusion criterion. Prediabetes was defined as fasting glucose concentration of 5.6 to 7.0 mmol/L and/or 2 h glucose concentration between 7.0 and 11.1 mmol/L by a 75 g oral glucose tolerance test (OGTT). Demographic and socioeconomic information (e.g., age, sex, and ethnicity), as well as lifestyle factors (e.g., smoking, alcohol consumption), were collected using standardized self-reported questionnaires [31,33]. Body mass index (BMI) was calculated as weight in kilograms divided by height in meters squared (kg/m^2^).

NMR spectra were acquired in the year 2000 using EDTA plasma samples from fasting participants collected at baseline (1992–1994), using NMR Profiler instruments at LipoScience (now Labcorp, Morrisville, NC, USA) [32,34]. Digitally stored NMR spectra were reanalyzed using the magnesium assay software. The sample size of the current study was 1342 participants after exclusion of subjects with missing NMR or covariate data or lack of information regarding diabetes status at baseline. The IRBs at each of the study sites approved the study protocol in 1991 and all participants provided written informed consent (NCT00005135). Venous EDTA plasma and serum samples were collected after an overnight fast and stored at <−70 °C until analysis. Glucose was measured shortly after blood sampling using the glucose oxidase technique on automated instruments (Yellow Springs glucose analyzer; YSI Inc., Yellow Springs, OH, USA). Lipids were analyzed according to Lipid Research Clinic methodology [32].

### 2.6. Statistical Analyses

Statistical analyses were performed using JMP version 12.1.0, Analyze-it v3.90.1 (Analyze-it Software, Ltd., Leeds, UK) or SAS v9.4 (SAS Institute, Cary, NC, USA). For the method comparison study, Deming and Passing–Bablok regression analyses were performed and residuals were evaluated using a Bland–Altman plot. For the epidemiological studies, data are expressed in mean ± SD. Skewed variables were natural log (Ln) transformed. Between-group differences in continuous variables were determined by ANOVA with subsequent unpaired *t*-tests in case of significant differences between the groups. Between-group differences in dichotomous variables were determined by chi-square analysis. Multivariable logistic regression analyses were performed to determine the independent associations of magnesium with diabetes status cross-sectionally and longitudinally. Stepwise adjustments were made for a priori selected potential confounders including age, sex, and race, BMI, fasting insulin, free fatty acids, GlycA (marker of systemic inflammation), LP-IR (lipoprotein-based measure of insulin resistance), lipids (total cholesterol, triglycerides, and HDL-cholesterol), HOMA-IR (glucose and insulin-based measure of insulin resistance) and fasting plasma glucose. In addition, interactions by age, sex, race, BMI, fasting glucose, fasting insulin, free fatty acids, total cholesterol, triglycerides, GlycA, LP-IR, and HOMA-IR were examined. Results are expressed in odds ratios (OR) per 1 SD increment in plasma magnesium with 95% confidence intervals (CI). Two-sided *p*-values < 0.05 were considered statistically significant (except for the comparison between non-diabetic, pre-diabetic, and diabetic individuals in which case we applied Bonferroni correction, yielding a significance threshold of 0.025).

## 3. Results

### 3.1. Quantification of Ionized Magnesium Using an NMR-Based Deconvolution Algorithm

Concentrations of ionized magnesium were determined by modeling the single NMR peak attributed to the protons from EDTA-complexed magnesium (MgEDTA) in plasma (Figure 1). The NMR signal for EDTA-complexed magnesium was modeled using derived components that, when combined, mimic the lineshape (i.e., shape of the line comprising the NMR signal) using a technique called deconvolution. The magnesium concentration was determined by converting the area of the signal peak into concentration units (µM) using a conversion factor determined from a standard curve with known magnesium concentrations.

### 3.2. Assay Characteristics and Stability of Magnesium in Plasma

Accuracy was determined through recovery experiments performed between 0 and 2008 µM magnesium (Table 1). To evaluate linearity, a regression analysis was performed on the magnesium test results versus the expected concentrations. The equation for the best line was determined to be Y = 0.956X + 10.6 with an R^2^ value of 0.999. Linearity was demonstrated over a large range of magnesium concentrations from 15.2 to 2715 µM. The LOB, LOD, and LOQ were determined to be: 0.0, 10.0, and 58.3 µM, respectively. As a result, the measuring range for the ionized magnesium assay was 58.3 to 2715 µM. Intra-assay (within-run) and inter-assay (within-laboratory) precision were evaluated and the results are summarized in Table 2. The %CV ranged from 0.7–1.5% for intra-assay, and 4.2–4.7% for inter-assay precision.

The stability of magnesium, when plasma samples were stored at different temperatures and after multiple freeze–thaw cycles, was evaluated. Magnesium concentrations were stable for up to 8 days when stored at ambient room temperature (20 to 25 °C), 15 days when refrigerated (2 to 8 °C), and 15 days when frozen at −20 °C. Stability was observed for up to 9 years when samples were stored <−70 °C. No significant change in results was observed after five freeze–thaw cycles.

The reference intervals for the NMR-based ionized magnesium assay were determined in plasma samples collected from a population of apparently healthy individuals. The mean ionized magnesium concentration was 640 ± 62.1 µM and the reference interval determined for the total population was 513 to 762 µM (Table 3).

### 3.3. Comparison between Results Generated by the NMR-Based Ionized Magnesium Assay in EDTA Plasma and Roche Total Magnesium Assay in Serum

Because most assays for measuring ionized magnesium are cumbersome and need specialized equipment, results generated by the NMR-based ionized magnesium assay were compared to those generated by a Roche total magnesium assay which is readily available in the clinical laboratory (Figure 2A). In a population of 5040 individuals from the PREVEND study, the NMR-based assay generated a mean ionized magnesium concentration of 667 ± 54 µM while the Roche total magnesium assay produced a mean concentration of 826 ± 56 µM. There was a direct correlation between the two measurements (correlation coefficient = 0.77), and as expected, the results generated by the NMR method were systematically lower than those produced by the total magnesium assay (Passing–Bablok fit: Y = 0.9816x − 0.1426). The Bland–Altman plot revealed that the residuals were equally distributed around the mean (Figure 2B).

### 3.4. Association of Ionized Magnesium with T2D in a Fairly High-Risk Study Population

Demographics and clinical characteristics of the study population, divided by subjects without diabetes (n = 614), with prediabetes (n = 301), and with T2D (n = 427) at baseline, are listed in Table 4. Compared to subjects without diabetes at baseline, those who had prediabetes or T2D were more likely to be men. Those with prediabetes and T2D were also more likely to have a higher BMI as well as higher fasting glucose, insulin, free fatty acids, triglycerides, GlycA, HOMA-IR, and LP-IR and lower HDL-cholesterol (Table 4). In addition, subjects with prediabetes had higher total cholesterol than non-diabetic participants at baseline, and those who had T2D were more likely to be of African American descent. However, total cholesterol levels did not differ significantly between subjects without diabetes and those with T2D. Compared to subjects with prediabetes, those who had T2D at baseline were more likely to have higher BMI, fasting glucose, insulin, free fatty acids, triglycerides, HOMA-IR, and LP-IR and lower HDL cholesterol. At baseline, NMR-measured ionized magnesium concentrations were significantly lower in subjects with prediabetes and T2D than subjects without diabetes and were significantly lower in subjects with T2D than in subjects with prediabetes (Table 4).

A cross-sectional analysis (n = 1342) revealed that ionized magnesium was independently associated with T2D at baseline in a logistic regression model adjusted for age, sex, and race (*p* < 0.0001) as well as in a model that was further adjusted for BMI, fasting insulin, free fatty acids, and GlycA (*p* < 0.0001) (Table 5). Moreover, the addition of LP-IR did not attenuate the association (both *p* < 0.0001). Only the addition of HOMA-IR attenuated the association between ionized magnesium and diabetes status (Table 5); however, the association remained significant (*p* = 0.032). No interactions were noted between ionized magnesium and age, race, BMI, fasting glucose, fasting insulin, free fatty acids, total cholesterol, triglycerides, GlycA, or LP-IR. However, there was a significant interaction between magnesium and sex (*p* = 0.0028) as well as HOMA-IR (*p* = 0.046). Therefore, regression analyses were performed separately in men and women. In men (n = 673), HOMA-IR (*p* = 0.90) fully attenuated the association. However, in women (n = 669), the association remained significant even adjusting for HOMA-IR (*p* = 0.0015) (Table 5).

Of the original 1342 subjects, 915 were without diabetes at baseline, of which 82 were lost to follow-up or lacked NMR data. Of the 833 remaining subjects, 131 were newly diagnosed with T2D by the 5 year follow-up visit; 79 were men and 52 were women. Subjects who were diagnosed with T2D tended to have lower baseline ionized magnesium (n = 131; 617 ± 133 µM) than those who remained free of diabetes throughout the study (n = 702; 638 ± 118 µM) (*p* = 0.067). Because there were no available time-to-event data for the IRAS study, logistic regression analyses were employed to investigate the potential association of ionized magnesium with incident T2D at 5 years (Table 6). Ionized magnesium was significantly associated with a decreased risk of developing T2D in a model adjusted for age, sex, and race (*p* = 0.035). The association remained independent after further adjustment for BMI, fasting insulin, free fatty acids, and inflammation as measured by GlycA as well as HOMA-IR (*p* = 0.033). However, the association of ionized magnesium with future T2D was attenuated by the addition of LP-IR (*p* = 0.22) to the model (Table 6). No significant interactions were noted between ionized magnesium and age, race, BMI, fasting glucose, fasting insulin, free fatty acids, total cholesterol, triglycerides, GlycA, LP-IR, or HOMA-IR. Similar to the cross-sectional analysis, a significant interaction between magnesium and sex (*p* = 0.0039) was found in the longitudinal analysis. However, the association of ionized magnesium with incident T2D showed only a trend for an association in men and absence of an association in women when the total group was divided by sex (Table 6).

## 4. Discussion

Given the reported associations between ionized magnesium levels in multiple chronic diseases [1], we developed a high-throughput NMR-based assay for determining plasma ionized or free magnesium concentrations in the clinical laboratory. The assay was optimized to improve its accuracy and precision and re-evaluated for its performance. The present study showed that the current NMR-based assay was robust and had performance characteristics consistent with the use of this assay in the clinical laboratory. To extend the previous observations in a north-European general population (PREVEND cohort study) [11], the current study investigated the cross-sectional and longitudinal associations of ionized magnesium with T2D in a population of ethnically diverse subjects who were chosen based on their glycemic status. The results revealed that lower ionized magnesium concentrations were associated with T2D at baseline in the total study population as well as separately in women, but not in men. In agreement with the previously published study, lower ionized magnesium concentrations were associated with incident T2D in IRAS, a study in subjects at higher risk of progressing to T2D based on the fact that many of these subjects had prediabetes at the baseline of the study. The association between magnesium and future T2D was attenuated by LP-IR. In contrast to the results of the cross-sectional analyses, the association was neither significant in women nor in men when sexes were analyzed separately. This could be due to the fact that there were fewer cases of future T2D when the cohort was divided by sex. These study results confirmed that ionized magnesium measurements are clinically useful for assessing a patient’s risk of progressing to T2D, even in a diverse population with varying glucose levels.

Multiple studies have reported associations between serum or plasma magnesium and risk of developing T2D [11,35,36,37]. Two studies reported an inverse association with lower circulating magnesium concentrations being related to a higher risk of developing T2D in a general population [11,36]. Kieboom et al. reported that serum magnesium was associated with prediabetes and revealed that the association was attenuated by the addition of insulin resistance, as determined by HOMA-IR [38]. In the present study, the association of NMR-measured ionized magnesium with incident T2D was not attenuated by HOMA-IR but was attenuated by LP-IR, a measure of insulin resistance that is dependent on changes in lipoprotein parameters that typically occur in subjects with metabolic disease [16]. These observations support the results of the Kieboom study and further support the notion that the association between magnesium and risk of prediabetes and T2D may be partly mediated by insulin resistance and is associated with the presence of diabetic dyslipidemia (high triglycerides and low HDL cholesterol). It has been reported previously that insulin resistance, as measured by LP-IR, is associated with future T2D and is an early indicator of progression to T2D given that it can predict T2D even in subjects with normal glucose levels [16,23]. Whether the lower magnesium concentrations are caused by insulin resistance or are a reflection of common underlying pathophysiological issues is unknown at this time and warrants further investigation. However, in a previous report among PREVEND participants, we found that plasma magnesium is more closely related to hyperglycemia than to hypertriglyceridemia [12]. Future studies measuring ionized magnesium using this NMR-based assay may be needed to further understand the observed associations.

Ionized or free magnesium represents 55–70% of circulating total magnesium concentrations [1,2,3,4]. In the PREVEND study, NMR-measured ionized magnesium was about 80% that of the total circulating magnesium concentration. This suggests that besides free magnesium, the EDTA in the sample and assay buffer may be chelating free magnesium as well as magnesium that is loosely bound to citrates, phosphates, or proteins. With that said, the previously reported range for serum ionized magnesium in healthy adults was 0.55–0.75 mM [1] which is in line with the reference interval that was obtained in apparently healthy individuals in the current study (0.51–0.76 mM). This supports the fact that the NMR assay is measuring chelatable ionized or free magnesium and is not a measure of total magnesium where the circulating concentration is significantly higher 0.70–1.05 mM [9,10]. Previously, Greenway et al. measured ionized magnesium photometrically in whole blood using a NOVA8 electrolyte analyzer and reported that the ionized to total magnesium concentration had a correlation coefficient (r) of 0.60 [39]. Moreover, in a recent article by Rooney et al. the authors measured ionized magnesium in whole blood using a pHOx^®^ Ultra blood gas analyzer and reported a correlation coefficient (r) of 0.50 [40]. In the current study, the correlation coefficient (r) in EDTA plasma was 0.77, confirming the fact that ionized magnesium and total magnesium concentrations do not correlate well and the level of correlation may depend on the study population, the technique used to quantify magnesium, and the type of specimen tested [39,40].

In addition to measuring the bioactive form of plasma magnesium, the NMR assay is high-throughput and easy to use and is therefore amenable for use in testing samples from large observational and interventional clinical studies. Most assays for measuring ionized magnesium are cumbersome and need specialized equipment; therefore, a simple high-throughput assay, especially for large clinical trials, is warranted. The stability of ionized magnesium in the specimen to be tested is also a factor for testing of clinical trial samples. Issues with the stability of ionized magnesium in whole blood samples were reported [40]. Compared to fresh whole blood, ionized magnesium was higher in refrigerated and frozen samples [40]. In the current study, we interrogated stability and found that ionized magnesium, measured in EDTA plasma samples, was extremely stable when stored at ambient room temperature (20 to 25 °C), refrigerated (2 to 8 °C), and for 15 days when frozen at −20 °C. Furthermore, stability was observed for up to 9 years when samples were stored <−70 °C. This suggests that measuring ionized magnesium in fresh or frozen EDTA plasma samples via NMR is highly suitable for measurement in clinical trial samples. Previous clinical trials have shown that supplementation with magnesium may be effective in reducing fasting plasma glucose in patients with T2D and HOMA-IR in individuals at risk of T2D [35]. Future clinical studies with magnesium supplementation, measurement of EDTA plasma samples using NMR, and interrogation of the relationship between ionized magnesium with disease outcomes are warranted.

The strengths of this study include the diversity of the IRAS participants and the fact that they had varying glucose levels at study start, including subjects with prediabetes; patients whose risk of progressing to T2D were most frequently assessed. In addition, the NMR-based assay is high-throughput, accurate, and measures ionized magnesium, which is thought to be the more bioactive form of magnesium. Limitations of this study should be considered including the fact that cause–effect relationships could not be ascertained and residual confounding may exist.

## 5. Conclusions

Lower ionized magnesium concentrations were associated cross-sectionally with T2D. In addition, ionized magnesium was moderately associated with incident T2D in IRAS, a study in ethnically diverse subjects at higher risk of progressing to T2D.

## Figures and Tables

**Figure 1 nutrients-14-01792-f001:**
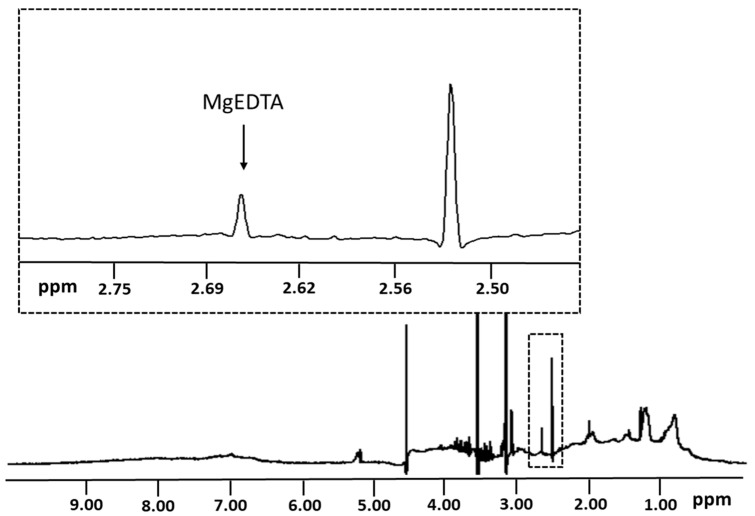
1D 1H NMR spectrum for EDTA-anticoagulated plasma collected on a 400 MHz spectrometer (see text) showing the peak corresponding to ethylene protons (i.e., -N-CH_2_-CH_2_-N-) from EDTA-complexed magnesium. This peak (labeled as MgEDTA) was used to quantify ionized magnesium in the sample.

**Figure 2 nutrients-14-01792-f002:**
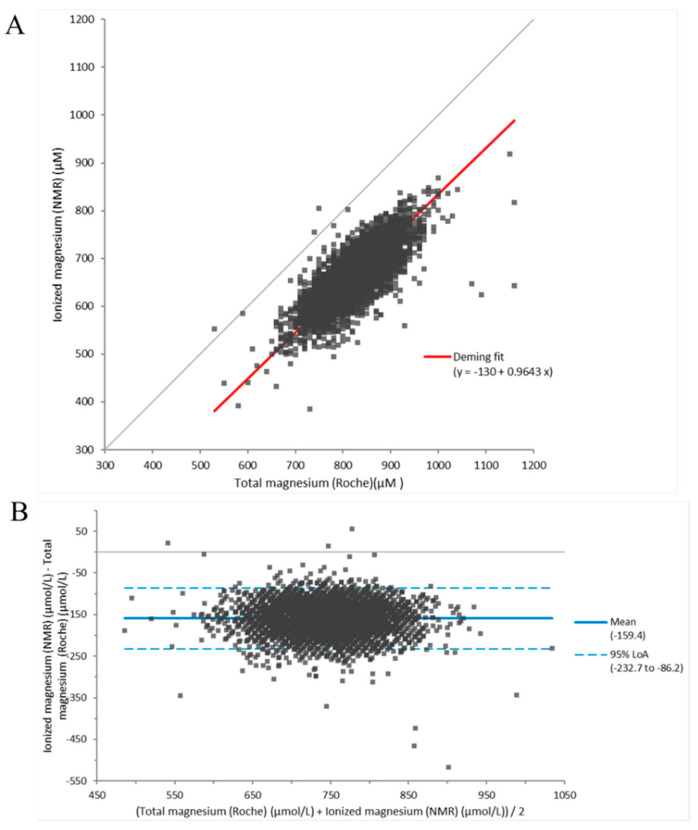
Method comparison between ionized magnesium and total magnesium assays in the Prevention of End-Stage Renal Disease (PREVEND) study (n = 5040). (**A**) Deming regression analysis, and (**B**) Bland–Altman Plot.

**Table 1 nutrients-14-01792-t001:** Assessment of accuracy of the NMR magnesium assay.

Sample #	Spiked (µM)	Measured ^a^ (µM)	Recovered (µM)	% Recovery
1	0	589.9 ± 12.8 ^b^	--	--
2	105.1	688.5 ± 12.6	98.6	93.8
3	304.9	883.0 ± 9.5	293.1	96.1
4	609.8	1156.4 ± 5.2	566.5	92.9
5	809.5	1330.0 ± 12.0	740.1	91.4
6	1009.3	1544.1 ± 9.8	954.2	94.5
7	1503.4	2003.6 ± 41.8	1413.8	94.0
8	2008.1	2479.1 ± 36.9	1889.3	94.1

# number. ^a^ mean of four replicates ± standard deviation (SD). ^b^ measured magnesium before spiking.

**Table 2 nutrients-14-01792-t002:** Within-run and within-laboratory imprecision.

	Magnesium (µM)		
	Low	Intermediate	High
**Within-Run** ^a^			
Mean	561	799	1222
SD	8	6	11
%CV	1.5	0.7	0.9
**Within-Laboratory** ^b^			
Mean	567	762	1179
SD	25	36	50
%CV	4.5	4.7	4.2

^a^ Based on 1 run of 20 tests (n = 20), ^b^ Based on 2 runs per day in duplicate for 20 days (n = 80).

**Table 3 nutrients-14-01792-t003:** Distribution of magnesium in EDTA plasma in apparently healthy adults (n = 564).

Percentile	Value (µM)
Min	433
0.5th	480
2.5th	513
10th	555
25th	598
50th	644
75th	681
90th	717
97.5th	762
99.5th	796
Max	843
**Mean**	640
**SD**	62.1
**95% Reference Interval**	513–762

**Table 4 nutrients-14-01792-t004:** Baseline demographics and clinical characteristics of subjects without diabetes, with prediabetes, and with type 2 diabetes in the Insulin Resistance Atherosclerosis Study (IRAS) (n = 1342).

	No Diabetes (n = 614)	Prediabetes (n = 301)	Type 2 Diabetes (n = 427)	*p*-Value
Age (years)	54 ± 9 ^a,b^	57 ± 8	57 ± 8	<0.0001
Sex, men (%)	52 ^d^	41 ^f^	54	0.0015
Race				
Non-Hispanic white (%)	41	39	34	0.096
Hispanic (%)	33	34	32	0.79
African American (%)	26 ^e^	27	34	0.015
BMI (kg/m^2^)	27.4 ± 4.9 ^a,b^	30.3 ± 6.2	31.3 ± 5.6	<0.0001
Fasting glucose (mg/dL)	96 ± 10 ^d,b^	105 ± 11 ^c^	175 ± 59	<0.0001
Fasting insulin (mIU/L)	13.8 ± 9.5 ^a,b^	19.7 ± 21.4 ^c^	23.3 ± 16.5	<0.0001
Fasting FFA (mmol/L)	0.43 ± 0.17 ^a,b^	0.55 ± 0.19 ^f^	0.59 ± 0.23	<0.0001
Total cholesterol (mg/dL)	208 ± 44 ^d^	216 ± 39	212 ± 43	0.033
Triglycerides (mg/dL)	125 ± 83 ^d,b^	159 ± 96 ^f^	189 ± 165	<0.0001
HDL-C (mg/dL)	47.2 ± 15.4 ^b^	45.0 ± 14.5 ^c^	40.0 ± 11.5	<0.0001
GlycA (µmol/L)	350 ± 64 ^a,b^	379 ± 76	381 ± 70	<0.0001
HOMA-IR	3.3 ± 2.4 ^a,b^	5.1 ± 5.9 ^c^	9.9 ± 7.9	<0.0001
LP-IR score (0–100)	41 ± 21 ^a,b^	49 ± 20 ^c^	56 ± 19	<0.0001
Ionized magnesium (µM)	644 ± 119 ^d,b^	612 ±123 ^c^	572 ± 135	<0.0001

Data are in mean ± SD for continuous or % for dichotomous variables. Between-group differences were determined by ANOVA with the Bonferroni method being applied to correct for multiple comparisons; *p*-value was calculated by ANOVA. ^a^
*p* < 0.0001 compared non-diabetic and pre-diabetic subjects; ^b^
*p* < 0.0001 compared diabetic and pre-diabetic subjects; ^c^
*p* < 0.0001 compared non-diabetic and diabetic subjects; ^d^
*p* < 0.05 compared non-diabetic and pre-diabetic subjects; ^e^
*p* < 0.05 compared non-diabetic and diabetic subjects; ^f^
*p* < 0.05 compared diabetic and pre-diabetic subjects; Abbreviations: BMI, body mass index; FFA, free fatty acids; HDL-C, high-density lipoprotein cholesterol; LP-IR, Lipoprotein Insulin Resistance Index. Triglycerides and insulin values were log transformed.

**Table 5 nutrients-14-01792-t005:** The association of NMR-measured ionized magnesium with prevalence of type 2 diabetes at baseline in Insulin Resistance Atherosclerosis Study (IRAS) participants (n = 1342).

	Total Participants (n = 1342)	Wald χ^2^	*p*-Value	Women (n = 669)	Wald χ^2^	*p*-Value	Men (n = 673)	Wald χ^2^	*p*-Value
Prevalent T2D, n (%)	427 (31.8)	-	-	196 (29.3)	-	-	231 (34.3)	-	-
Model 1	0.592 (0.523–0.671)	67.4084	<0.0001	0.479 (0.396–0.579)	57.5952	<0.0001	0.711 (0.602–0.841)	15.8831	<0.0001
Model 2	0.644 (0.562–0.738)	40.3684	<0.0001	0.537 (0.438–0.657)	36.0651	<0.0001	0.749 (0.623–0.901)	9.3833	0.0022
Model 3	0.638 (0.557–0.731)	41.7936	<0.0001	0.535 (0.437–0.656)	36.3385	<0.0001	0.737 (0.612–0.888)	10.3900	0.0013
Model 4	0.689 (0.599–0.792)	27.4432	<0.0001	0.606 (0.491–0.748)	21.8372	<0.0001	0.768 (0.636–0.927)	7.5675	0.0059
Model 5	0.771 (0.608–0.978)	4.6083	0.032	0.540 (0.369–0.789)	10.1375	0.0015	0.981 (0.714–1.349)	0.0139	0.90

Data are reported as odds ratio (95% confidence intervals). Abbreviations: BMI, body mass index; FFA, free fatty acids; HOMA-IR, homeostatic model assessment for insulin resistance; LP-IR, Lipoprotein Insulin Resistance Index. Model 1: Adjusted for age, sex, race. Model 2: Model 1 and BMI, fasting insulin, and FFA. Model 3: Model 2 and GlycA. Model 4: Model 3 and LP-IR. Model 5: Model 3 and HOMA-IR.

**Table 6 nutrients-14-01792-t006:** The association of NMR-measured ionized magnesium with the risk of developing type 2 diabetes at 5 year follow up visit in Insulin Resistance Atherosclerosis Study (IRAS) participants (n = 833). Insulin Resistance Atherosclerosis Study (IRAS) participants (n = 833).

	Total Participants (n = 833)	Wald χ^2^	*p*-Value	Women (n = 473)	Wald χ^2^	*p*-Value	Men (n = 360)	Wald χ^2^	*p*-Value
Events, n (%)	131 (15.7)	-	-	79 (16.7)	-	-	52 (14.4)	-	-
Model 1	0.817 (0.677–0.986)	4.4439	0.035	0.851 (0.666–1.087)	1.6731	0.20	0.772 (0.576–1.036)	2.9707	0.085
Model 2	0.814 (0.669–0.991)	4.1968	0.041	0.857 (0.662–1.110)	1.3614	0.24	0.739 (0.543–1.007)	3.6630	0.056
Model 3	0.808 (0.664–0.985)	4.4657	0.035	0.845 (0.651–1.097)	1.5988	0.21	0.739 (0.542–1.006)	3.6832	0.055
Model 4	0.802 (0.655–0.983)	4.5351	0.033	0.835 (0.637–1.093)	1.7252	0.19	0.734 (0.536–1.005)	3.7231	0.054
Model 5	0.881 (0.719–1.080)	1.4908	0.22	1.005 (0.760–1.329)	0.0010	0.97	0.745 (0.545–1.018)	3.4160	0.065

Events = type 2 diabetes at 5 year follow-up visit. Data are reported as odds ratio (95% confidence intervals) per 1 SD increment in plasma magnesium. Abbreviations: BMI, body mass index; FFA, free fatty acids; HOMA-IR, homeostatic model assessment for insulin resistance; LP-IR, Lipoprotein Insulin Resistance Index. Model 1: Adjusted for age, sex, race. Model 2: Model 1 and BMI, fasting insulin, and FFA. Model 3: Model 2 and GlycA. Model 4: Model 3 and HOMA-IR. Model 5: Model 3 and LP-IR.

## Data Availability

Data are available upon request.

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
