# Peer review of "Nuclear Magnetic Resonance-Measured Ionized Magnesium Is Inversely Associated with Type 2 Diabetes in the Insulin Resistance Atherosclerosis Study"

_nutrients, 2022, doi:10.3390/nu14091792_

Round 1

Reviewer 1 Report

I congratulate the authors for their beautyful and interesting work.

Here are my suggestion to improve the paper:

  1. lines 49-54 Please insert metabolic syndrome. here a reference: Piuri, G.; Zocchi, M.; Della Porta, M.; Ficara, V.; Manoni, M.; Zuccotti, G.V.; Pinotti, L.; Maier, J.A.; Cazzola, R. Magnesium in Obesity, Metabolic Syndrome, and Type 2 Diabetes. Nutrients 2021, 13, 320. https://doi.org/10.3390/nu13020320
  2. line 64, please insert the full term of the acronym "GlycA" and briefly detail better what is GlycA.
  3. Line 99 please better explain what is LP-IR?
  4. TABLE 1,  insert the SD in the values of the column of Measured Magnesium.
  5. After line 350, please indicate that the association of ionized magnesium and the adjustment for entire parameters is not significant when the total group was divided by sex.
  6. Lines 385-388 the association between magnesium and future T2D was not significant when the group was divided by sex. Indicate this and try to find an explanation.

Author Response

Thank you for your kind words.  We appreciate your comments and suggestions as they have improved our manuscript.  Please find our comments below.

  1. lines 49-54 Please insert metabolic syndrome. here a reference: Piuri, G.; Zocchi, M.; Della Porta, M.; Ficara, V.; Manoni, M.; Zuccotti, G.V.; Pinotti, L.; Maier, J.A.; Cazzola, R. Magnesium in Obesity, Metabolic Syndrome, and Type 2 Diabetes. Nutrients 2021, 13, 320. https://doi.org/10.3390/nu13020320

Thank you for pointing out this interesting paper.  We agree that it is relevant to the introduction and have added “metabolic syndrome” to line 52 and have added the reference on line 54 (reference 14).

  1. line 64, please insert the full term of the acronym "GlycA" and briefly detail better what is GlycA.

GlycA is the full name (not an abbreviation) given to the NMR signal that arises largely from the N-acetyl glucosamine residues on the side chains of acute phase glycoproteins. We have added an additional sentence and references that explain what GlycA is in more detail (lines 65-68; references 25-27).  

  1. Line 99 please better explain what is LP-IR?

So as not to get lost in the methods section, we added a sentence and references about LP-IR to the introduction (line 65 and lines 68-70; references 16 and 28).

  1. TABLE 1, insert the SD in the values of the column of Measured Magnesium.

We have added the SD values to the column of measured magnesium in Table 1.

  1. After line 350, please indicate that the association of ionized magnesium and the adjustment for entire parameters is not significant when the total group was divided by sex.

We added the following sentence to the results section (lines 353-355). “However, the association of ionized magnesium with incident T2D showed only a trend for an association in men and absence of an association in women when the total group was divided by sex (Table 6).

  1. Lines 385-388 the association between magnesium and future T2D was not significant when the group was divided by sex. Indicate this and try to find an explanation.

The following sentence is in the discussion section “In contrast to the results of the cross-sectional analyses, the association was neither significant in women nor in men, when sexes were analyzed separately.” We also added the following potential explanation “This could be due to the fact that there were fewer cases of future T2D when the cohort was divided by sex.”

Thank you for your kind words.  We appreciate your comments and suggestions as they have improved our manuscript.  Please find our comments below.

  1. lines 49-54 Please insert metabolic syndrome. here a reference: Piuri, G.; Zocchi, M.; Della Porta, M.; Ficara, V.; Manoni, M.; Zuccotti, G.V.; Pinotti, L.; Maier, J.A.; Cazzola, R. Magnesium in Obesity, Metabolic Syndrome, and Type 2 Diabetes. Nutrients 2021, 13, 320. https://doi.org/10.3390/nu13020320

Thank you for pointing out this interesting paper.  We agree that it is relevant to the introduction and have added “metabolic syndrome” to line 52 and have added the reference on line 54 (reference 14).

  1. line 64, please insert the full term of the acronym "GlycA" and briefly detail better what is GlycA.

GlycA is the full name (not an abbreviation) given to the NMR signal that arises largely from the N-acetyl glucosamine residues on the side chains of acute phase glycoproteins. We have added an additional sentence and references that explain what GlycA is in more detail (lines 65-68; references 25-27).  

  1. Line 99 please better explain what is LP-IR?

So as not to get lost in the methods section, we added a sentence and references about LP-IR to the introduction (line 65 and lines 68-70; references 16 and 28).

  1. TABLE 1, insert the SD in the values of the column of Measured Magnesium.

We have added the SD values to the column of measured magnesium in Table 1.

  1. After line 350, please indicate that the association of ionized magnesium and the adjustment for entire parameters is not significant when the total group was divided by sex.

We added the following sentence to the results section (lines 353-355). “However, the association of ionized magnesium with incident T2D showed only a trend for an association in men and absence of an association in women when the total group was divided by sex (Table 6).

  1. Lines 385-388 the association between magnesium and future T2D was not significant when the group was divided by sex. Indicate this and try to find an explanation.

The following sentence is in the discussion section “In contrast to the results of the cross-sectional analyses, the association was neither significant in women nor in men, when sexes were analyzed separately.” We also added the following potential explanation “This could be due to the fact that there were fewer cases of future T2D when the cohort was divided by sex.”

Reviewer 2 Report

The aims of this paper, as the authors note, were to improve the precision and accuracy of this magnesium assay by assay optimization, assess assay performance, and determine the longitudinal association of NMR-measured plasma ionized magnesium with future T2D in the IRIS study cohort.  

The authors were extremely meticulous in describing the methods of NMR - measured plasma ionized magnesium. They sought to optimize their methodology and evaluate the precision, and short term as well as long term stability of collected samples for the magnesium assay.  They also performed a comparison study of use of standard colorimetric methodology for serum magnesium and compared to the NMR method.  Reference intervals were discussed for both standard colorimetric and the NMR method with a direct correlation noted between both methods. 

Much debate has taken place regarding an optimal range for serum magnesium with varying cut points for the low range so as to be able to identify individuals that may be at risk for chronic latent magnesium deficiency.  These investigators have demonstrated that the NMR method tested in samples from populations at risk for diabetes mirrors the results reported in the literature for associations of low serum magnesium and development of metabolic syndrome and diabetes.  

The magnesium field lacks a robust biomarker for magnesium assessment and the use of NMR-based ionized magnesium may provide the needed validation of the role of magnesium and chronic disease. To establish a population based reference interval greater numbers of subjects, all ages, will be required and the methodology must be available to not only large research universities but community hospitals as well.

Can the investigators comment on the cost per sample using the NMR technology and what the start up costs might be for a hospital laboratory?

The authors may want to work with the National Institute of Standards and Technology (NIST) to develop a reference standard for the NMR methodology. 

Author Response

Thank you for your kind words.  We appreciate your comments and suggestions as they have improved our manuscript.  Please find our comments below.

  1. Can the investigators comment on the cost per sample using the NMR technology and what the start up costs might be for a hospital laboratory?

Unfortunately, the NMR-based assay is only available for measuring ionized magnesium in clinical trial samples.  The current cost per clinical trial sample ranges from $54 to $80 depending on the number of samples to be tested, due to the cost of sample and study management. While the NMR technology using Vantera Clinical Analyzers is currently available in a large clinical laboratory in the United States, it is not likely to be placed in the hospital setting. Benchtop NMR instruments are being evaluated and may be suitable for these purposes in the future.

  1. The authors may want to work with the National Institute of Standards and Technology (NIST) to develop a reference standard for the NMR methodology.

This is a great idea and one that we will pursue in the coming months/years.